# Double and Single Descent in Causal Inference with an Application to High-Dimensional Synthetic Control

**Jann Spiess**     **Guido Imbens**     **Amar Venugopal**
Stanford University
Stanford, CA, USA
`jspiess,imbens,amarvenu@stanford.edu`

## Abstract

Motivated by a recent literature on the double-descent phenomenon in machine learning, we consider highly over-parameterized models in causal inference, including synthetic control with many control units. In such models, there may be so many free parameters that the model fits the training data perfectly. We first investigate high-dimensional linear regression for imputing wage data and estimating average treatment effects, where we find that models with many more covariates than sample size can outperform simple ones. We then document the performance of high-dimensional synthetic control estimators with many control units. We find that adding control units can help improve imputation performance even beyond the point where the pre-treatment fit is perfect. We provide a unified theoretical perspective on the performance of these high-dimensional models. Specifically, we show that more complex models can be interpreted as model-averaging estimators over simpler ones, which we link to an improvement in average performance. This perspective yields concrete insights into the use of synthetic control when control units are many relative to the number of pre-treatment periods.

## 1   Introduction

Motivated by a recent literature on the double-decent phenomenon in machine learning, we investigate the properties of common econometric estimators when we increase their complexity. For high-dimensional linear regression and for synthetic control with many control units, we document empirical applications where extremely over-parameterized models impute missing out-of-sample and out-of-time outcomes well. We then provide a common explanation for the returns to complexity in high-dimensional linear regression and synthetic control in terms of a simple model-averaging property, which we link to an improvement in imputation performance.

We often conceptualize the effect of complexity on econometric models in terms of a bias–variance trade-off: Adding complexity makes models more expressive and reduces bias, while increased overfitting leads to additional variance. In this view, an overly complex model fits overly well in the training sample, but fails to recover true parameters or generalize to new data. For example, linear regression with too many variables or synthetic control with many control units may perform poorly because of overfitting and excess variance. Consistency results for high-dimensional econometric models therefore usually assume that the expressiveness of models is limited relative to the available sample size, and practical advice often highlights choosing simple models or regularizing complex ones, in a way that balances bias and variance optimally to achieve good estimation and prediction performance.

---

Replication code is available at [github.com/amarvenu/causal-descent](github.com/amarvenu/causal-descent).

37th Conference on Neural Information Processing Systems (NeurIPS 2023).

A recent literature in statistics and machine learning has highlighted the surprisingly good prediction performance of extremely high-dimensional models that fit the data perfectly. That literature has documented a so-called double-descent phenomenon for deep neural networks and other high-dimensional regression models: increasing complexity beyond the interpolation threshold at which the training sample error is zero can lead to a gradual reduction in variance and improvement in out-of-sample performance. In such cases, there are often two complexity regimes: below the interpolation threshold, the usual bias–variance trade-off leads to a decrease (first descent) and then an increase in out-of-sample loss, while beyond the interpolation threshold, out-of-sample loss decreases again (second descent).

In order to investigate the properties of highly over-parameterized models in causal inference, we first demonstrate a double-descent curve for linear regression in imputing wage outcomes and estimating average-treatment effects. In the LaLonde (1986) sample of the Current Population Survey (CPS) drawn from Dehejia and Wahba (1999, 2002), we generate over 8,000 variables from binning and interacting the original eight demographic and employment-related variables. We then fit a linear regression model on 3,000 training units and an increasing, randomly chosen subset of these variables, choosing the norm-minimizing solution once the model fits perfectly. Evaluated on the LaLonde (1986) control sample from the National Supported Work Demonstration (NSW) experiment, we observe the usual bias–variance trade-off for a low and moderate number of included covariates, where performance first increases slightly, before deteriorating substantially when approaching the interpolation threshold. Beyond the interpolation threshold, however, out-of-sample performance increases again. Ultimately, an extremely over-parameterized model on over 8,000 variables even outperforms less complex regressions with a randomly chosen set of covariates and achieves performance comparable to a linear regression on the original, unmanipulated set of covariates.

Having demonstrated the returns to complexity in high-dimensional linear regression, we document the performance of synthetic control estimators with many control units. In the California smoking data (Abadie et al., 2010), we impute missing smoking rates based on a small number of pre-treatment periods and an increasingly large number of control states. As in the case of linear regression, we see returns to increasing model complexity, even beyond the point where some of the synthetic-control models fit the training data perfectly. However, for synthetic control we do not observe an initial trade-off between bias and variance: in our empirical example, the performance on future periods is always better for the more complex models, with no intermittent increase in errors. Unlike the double-descent relationship for linear regression, the performance of synthetic control in our application yields a single-descent curve that only improves with complexity, no matter whether there are a few or many control states.

We then connect the returns to complexity in high-dimensional linear regression and in synthetic control in terms of a simple model-averaging property. While both estimators are constructed differently and represent different regressions, they both share a common feature: More complex models can be represented as convex averages over simpler models. We show that this model-averaging property applies to linear regression in the interpolation regime, as well as to synthetic control in general. The property holds purely mechanically and does not depend on the training or target distributions. It relates to other model-agnostic properties of linear regression, for which we also show a reduction in (conditional) variance for minimal-norm least-squares estimators beyond the interpolation threshold.

Having established a model-averaging property for interpolation linear regression and for synthetic control, we provide high-level assumptions under which this property translates to better imputation performance. A direct consequence of model averaging is that the (convex) prediction loss of a more complex model cannot be worse than the corresponding average prediction loss of simpler models, when the same weights are used to average. When the complex model on average also outperforms comparable models of the same complexity, then we show that the model-averaging property translates into a reduction in average out-of-sample error relative to a randomly selected simpler model. These results only put high-level, largely model-agnostic assumptions on the data-generating process, and are driven by mechanical properties of the underlying estimators.

Our results have practical implications for the use of synthetic control with many control units. Conventional wisdom may indicate that synthetic control with a very large number of donor units relative to the number of pre-treatment periods is problematic, and that selecting among many, ex-ante

exchangeable units in this case may represent a challenge. However, our results imply that this is not the case: as long as ties are broken by a suitable regularization procedure (such as the minimum-norm solution in our case), making an ex-ante choice among many control units is not necessary. That said, if ex-ante information is available about which units are particularly suitable as controls, then using this information can still be helpful and improve imputation.

We build upon results on over-parameterized regression in machine learning and statistics, where double-descent curves for kernel and linear regression and the performance of norm-minimizing interpolating solutions have been studied extensively (including Liang and Rakhlin, 2020; Liang et al., 2020; Liang and Recht, 2023; Bartlett et al., 2020; Hastie et al., 2022) as part of a broader literature on double descent and interpolation in deep learning (Zhang et al., 2016; Belkin et al., 2018; Belkin, 2021; Mei and Montanari, 2022). Kelly et al. (2022) documents the benefits of over-parameterized models in asset return prediction both in theory and empirically. Kato and Imaizumi (2022) provides results on the estimation of conditional average causal effects using over-parametrized linear regression. Relative to this work on interpolating regression, we show connections to synthetic control based on simple mechanical properties that are largely agnostic about the true data-generating process. Thereby, we also relate to work on high-dimensional and regularized synthetic control (Doudchenko and Imbens, 2016; Abadie and L'Hour, 2021; Ben-Michael et al., 2021) and the connections between synthetic control and linear regression (Athey et al., 2021; Agarwal et al., 2021; Shen et al., 2022; Bruns-Smith et al., 2023).[1] Finally, we connect to a literature on model averaging in econometrics and statistics (e.g. Hansen, 2007; Claeskens and Hjort, 2008). More specifically, Wilson and Izmailov (2020) consider Bayesian model averaging in deep learning, and discuss relationships to double descent. While our linear-regression solutions are closely related to available results on norm-minimizing regression, we are not aware that the results on synthetic control were noted previously.

The remaining note is structured as follows. Section 2 provides an empirical example of double descent for linear regression and discusses some properties of the norm-minimizing linear-regression estimator in the interpolating case. Section 3 discusses the relationship of complexity and imputation performance of synthetic control in an empirical example, and makes connections to the properties of interpolating linear regression. Section 4 discusses high-level consequences of the model-averaging property of interpolating linear regression and synthetic control. Section 5 concludes by discussing some limitations and open questions.

## 2 Double Descent for Linear Regression

In this section, we consider imputation by high-dimensional linear regression as an illustration of the performance of highly over-parameterized models. We start with an empirical illustration of wage imputation in CPS data with a varying number of randomly ordered covariates, which we evaluate in terms of its ability to estimate an average treatment effect. We then discuss theoretical properties of the interpolating linear-regression estimator, followed by a graphical illustration. These results are closely related to prior work on interpolating linear and kernel regression (including Bartlett et al., 2020; Liang et al., 2020; Hastie et al., 2022; Kato and Imaizumi, 2022).

### 2.1 Setup and Estimator

We consider a linear-regression estimator in data $(y_i, x_i)_{i=1}^n \in \mathbb{R} \times \mathbb{R}^k$ from $n$ observations of a scalar outcome and $k$ scalar covariates. We write $X = (x_i')_{i=1}^n \in \mathbb{R}^{n \times k}$ and $Y = (y_i)_{i=1}^n \in \mathbb{R}^n$. For a subset $J \subseteq \{1, \ldots, k\}$ of the covariates, we denote by $\mathcal{B}^J = \arg\min_{\beta \in \mathbb{R}^k; \beta_j = 0 \forall j \notin J} \sum_{i=1}^n (y_i - x_i'\beta)^2$ the set of all least-squares linear-regression estimates on $J$. Among these, we choose the norm-minimizing estimate $\hat{\beta}^J = \arg\min_{\beta \in \mathcal{B}^J} \|\beta\|$. Throughout, we assume that $X_J = (x_{ij})_{i \in \{1, \ldots, n\}, j \in J} \in \mathbb{R}^{n \times |J|}$ is of full row rank. Hence, there are multiple least-squares solutions, $|\mathcal{B}^J| > 1$, if and only if $|J| > n$. In that case, $Y = X\hat{\beta}^J$ and $\hat{\beta}^J$ is the minimal-norm interpolating solution (cf. Liang et al., 2020).

---

[1]For example, Bruns-Smith et al. (2023) shows that synthetic-control-type balancing estimators with non-negative weights can be related to outcome regressions, which connects our OLS setup in Section 2 to the synthetic-control setup in Section 3.

## 2.2 Empirical Motivation

We impute wages in the non-experimental LaLonde (1986) sample of the Current Population Survey (CPS) using linear regression on a large number of covariates, based on data drawn from Dehejia and Wahba (1999, 2002). From the original eight covariates, we obtain $k = 8408$ explanatory variables by binning and interacting the available features. We train minimal-norm least-squares regression models on a training sample of $n = 3000$ randomly chosen observations, and evaluate their mean-squared error in imputing average wages for subsets (of various sizes) of the 260 LaLonde (1986) National Supported Work Demonstration (NSW) experimental controls as provided through Dehejia and Wahba (1999, 2002). To these covariates we add a small amount of iid Gaussian noise to avoid rank deficiency when estimating linear regression. When fitting wage imputation models, we vary the set of covariates included in the regression. Specifically, we order all features randomly. For a complexity $\ell \in \{1, \ldots, k\}$, we then choose the first $\ell$ covariates, and obtain the estimate $\hat{\beta}^J$ for $J = \{1, \ldots, \ell\}$. Our presented results reflect an average over five such random orderings of covariates.

In order to assess the viability of this approach for applications to average treatment effect (ATE) estimation, we assess each model by its ability to accurately predict the average outcome on a new dataset. In particular, we consider various subsets of size $m$ of the NSW experimental control set. For each $m$, we draw 1000 samples of size $m$ without replacement from the NSW experimental control dataset. For each sample, we average the $m$ observation-level outcome predictions and compare the result to the true mean outcome of the given sample. We then take the RMSE across the 1000 draws of size $m$ to obtain our evaluation metric.[2]

In Figure 1a, we consider CPS data and plot the average root-mean-squared error (RMSE) for pointwise prediction of the wages, reporting the in-sample performance as well. The vertical dashed line denotes the interpolation threshold where $\ell = n$. Figure 1b shows the ATE RMSE metric, once again averaged over five random orderings of the covariates, for various subset sizes. The horizontal dashed lines show the RMSE of a simple linear regression on the original, unmanipulated (low-dimensional) features, colored according to corresponding sample size $m$. A zoomed-in version of Figure 1b focusing on the highly overparametrized regime can be found in Figure 7.

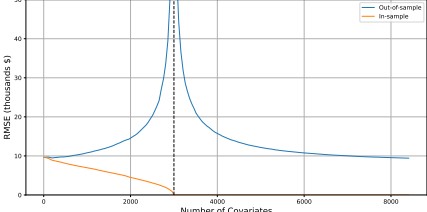

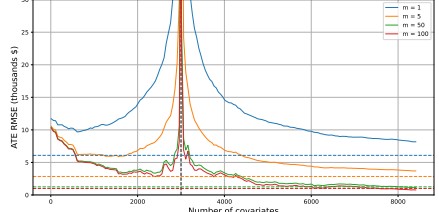

(a) Average out-of-sample (blue) and in-sample (orange) pointwise RMSE for CPS data.

(b) Average counterfactual prediction RMSE for varying subset sizes on NSW controls.

Figure 1: Average RMSE for linear regression for a varying number of covariates.

The out-of-sample losses in these illustrations show a descent in loss right of the interpolation threshold (denoted by the vertical dashed line), at which point in-sample loss is zero. For NSW experimental controls, out-of-sample error initially decreases (first descent), while for CPS non-experimental controls it remains initially flat. For both out-of-sample datasets, loss then goes on to increase and peaks sharply at $\ell = n$, at which point the linear models start to fit perfectly. As complexity increases further, loss decreases again (second descent). In both cases, loss continues to decrease throughout. For NSW experimental controls, loss ultimately reaches a minimum that is below the lowest error achieved left of the interpolation threshold, while the loss achieved in the CPS case is similar between the right tails and the minimum on the left. Furthermore, as the sample size $m$ increases in Figure 1b, the highly complex interpolating models outperform a simple model based

---

[2]For $m = 1$, taking 1000 draws would necessarily yield duplication; in this case, we instead simply consider the full set of 260 NSW experimental control observations. Note that results for $m = 1$ therefore correspond to a standard (observation-level) RMSE calculation, analogous to the out-of-sample curve in Figure 1a.

on the original set of provided features; the $m = 50$ and $m = 100$ curves have minima below their corresponding dashed lines.

In this setting, highly over-parameterized linear models constructed via discretizing and interacting available features exhibit performance improvements over a linear model fitted on the original features. This result appears remarkable, since the heavily over-parameterized models do not include the original non-binary covariates, but only indicators obtained from quantile binning (see Section B.1 for details). Further improvement could be achieved by always including the original covariates in the model.

We note that our illustration is extreme in that it artificially creates a large number of low-signal covariates, for which the best model only slightly outperforms a small, hand-curated model based on the original covariates. Nevertheless, this simple empirical exercise demonstrates the non-monotonicity in the relationship of complexity to variance and loss that has been studied by the literature on interpolating regression and double descent, and extends it to causal target parameters like the average treatment effect.

### 2.3 Theoretical Properties and Geometric Illustration

Motivated by the empirical illustration, we note some theoretical properties that are direct consequences of the structure of the norm-minimizing linear least-squares estimator, and will later serve as a comparison point for synthetic control. We note that formal results on the bias–variance properties in terms of more primitive properties of the data-generating process are available, including in Bartlett et al. (2020); Liang et al. (2020); Hastie et al. (2022). Here, we focus on illustrating properties that follow mechanically from the construction of the estimator. Our focus is on comparing more complex models, with covariates $J$, to slightly simpler (more sparse) ones, with covariates $J \setminus \{j\}$, where the $j$-th covariate is dropped. Throughout, we assume that the covariate matrices are of full row rank for the more and less complex models

**Assumption 1** (Full rank). *The covariate matrix $X_J \in \mathbb{R}^{n \times |J|}$ with columns $J$ as well as the covariate matrices $X_{J \setminus \{j\}} \in \mathbb{R}^{n \times (|J|-1)}$ for all $j \in J$ are of full row rank.*

We first consider the geometry of interpolating solutions, for which we note that more complex model can be expressed by an average over simpler models.

**Proposition 1** (Model averaging for interpolating linear least-squares regression). *For every $X$ and $J$ with $|J| > n$ such that Assumption 1 holds there exists*

$$\lambda \in [0,1]^J, \sum_{j \in J} \lambda_j = 1 \qquad \text{such that} \qquad \hat{\beta}^J = \sum_{j \in J} \lambda_j \hat{\beta}^{J \setminus \{j\}}.$$

We can choose the weights in Proposition 1 as a function of the covariate matrix $X_J$ only. Specifically, weights can be chosen as $\lambda_j = \frac{1 - X_j'(X_J X_J')^{-1} X_j}{|J| - n}$, where $X_j$ is the $j$-th column of $X$ and $X_j'(X_J X_J')^{-1} X_j$ can be seen as the "leverage" of feature $j \in J$, analogously to the leverage of an observation in the usual (low-dimensional) linear regression case.[3]

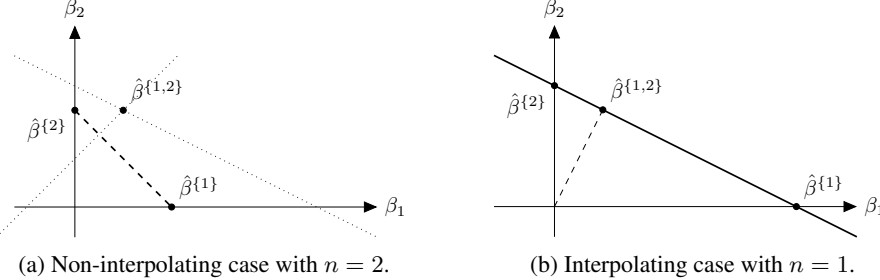

(a) Non-interpolating case with $n = 2$.      (b) Interpolating case with $n = 1$.

Figure 2: Minimal-norm least-squares solutions for linear regression with $J = \{1, 2\}$, where $n$ varies.

---

[3]We thank Tengyuan Liang for pointing out this connection to us, as well as for suggesting a direct proof via the Sherman–Morrison–Woodbury formula.

The model averaging property of interpolating linear regression is illustrated in the right panel of Figure 2 for the simple case of two covariates ($J = \{1, 2\}$) and a single data point ($n = 1$). Here, the models $\beta \in \mathbb{R}^J$ that for the model perfectly lie on the line through $\hat{\beta}^{\{1\}}$ and $\hat{\beta}^{\{2\}}$, with the norm-minimizing solution $\hat{\beta}^{\{1,2\}}$ lies between the two. In contrast, if $\hat{\beta}^{\{1\}}$ and $\hat{\beta}^{\{2\}}$ are not interpolating (such as in the case of the left panel of Figure 2, where $n = 2$), then the more complex model $\hat{\beta}^{\{1,2\}}$ does not generally lie in the convex hull of the simpler ones.

As an immediate consequence, any interpolating model can in this case be expressed as a convex average of simpler interpolating models, $\hat{\beta}^J = \sum_{L \subseteq J; |L| = \ell} \lambda_j \hat{\beta}^L$ for all $\ell \in \{n, \ldots, |J|\}$ (provided all $X_L$ are of full row rank). A particularly interesting special case is $\ell = n$, for which we express $\hat{\beta}^J$ as a model average of just-interpolating models $\hat{\beta}^L$ with $\hat{\beta}^L_L = X_L^{-1} Y$.

In addition to the model-averaging property, the geometry of interpolating solutions also implies that the variance of the norm-minimizing linear least-squares estimator generically decreases in the interpolation regime $|J| > n$, for which the complex estimator $\hat{\beta}^J$ along with the simpler estimator $\hat{\beta}^{J \setminus \{j\}}$ both fit the training data perfectly.

**Proposition 2** (Variance reduction for linear least-squares regression). *Suppose Assumption 1 and that Y has a second moment. If $|J| > n$ then a.s. $\operatorname{tr} \operatorname{Var}(\hat{\beta}^J | X) \leq \min_{j \in J} \operatorname{tr} \operatorname{Var}(\hat{\beta}^{J \setminus \{j\}} | X)$.*

The reduction in variance is a direct consequence of a more general property of the least-squares solution. Specifically, the next proposition shows that if we redraw new outcome data in the interpolation regime, then the distance between more complex solutions is smaller than the distance between less complex models.

**Proposition 3** (Variation hierarchy for linear least-squares regression). *For fixed X and J for which Assumption 1 holds consider two draws $Y_A$ and $Y_B$ yielding minimal-norm least-squares estimates $\hat{\beta}^J_A, \hat{\beta}^{J \setminus \{j\}}_A$ and $\hat{\beta}^J_B, \hat{\beta}^{J \setminus \{j\}}_B$, respectively.*

1. *If $|J| \leq n$, then $\|\hat{\beta}^J_A - \hat{\beta}^J_B\|_{X'X} \geq \max_{j \in J} \|\hat{\beta}^{J \setminus \{j\}}_A - \hat{\beta}^{J \setminus \{j\}}_B\|_{X'X}$.*

2. *If $|J| > n$, then $\|\hat{\beta}^J_A - \hat{\beta}^J_B\| \leq \min_{j \in J} \|\hat{\beta}^{J \setminus \{j\}}_A - \hat{\beta}^{J \setminus \{j\}}_B\|$.*

*Here, we write $\|\beta\|_M = \sqrt{\beta' M \beta}$ for some positive semi-definite symmetric matrix $M \in \mathbb{R}^{k \times k}$.*

In words, the variation of models across draws of the outcome data increases with complexity on the left of the interpolation threshold, while it decreases on the right. Here, the choice of norm is essential for these results to hold uniformly across simpler models.[4]

Figure 3 depicts this graphically. In Figure 3a, for the non-interpolating case we observe that the norm of the difference between the model coefficient vectors making use of both covariates ($\hat{\beta}^{\{1,2\}}, \tilde{\beta}^{\{1,2\}}$) is larger than the norm of the differences between the coefficient vectors taking into consideration just a single covariate at a time. In Figure 3b for the interpolating case, we observe that the reverse is true; in this case, the norm of the difference between the complex models is smaller than the norms of the differences between the simpler models.

In practice, we may care about model properties beyond variance, and consider imputation loss beyond the above norms in the parameters. In Section 4, we will leverage the model-averaging properties from Proposition 1 to establish such bounds on more general imputation errors.

We believe that these variance and geometric properties of linear regression are well understood in the literature and likely not new, although we are not aware of an explicit statement of the model-averaging connection between more and less complex interpolating linear-regression models.

## 3 Single Descent for Synthetic Control

We next consider imputation using synthetic control with many control units. As in the case of linear regression, we start with an empirical illustration. In the Abadie et al. (2010) California smoking

---

[4]The second result still holds for an alternative norm $\|\beta\|_M = \sqrt{\beta' M \beta}$ with $M$ positive definite and symmetric, provided that the same norm is used when selecting the norm-minimizing estimator in the interpolation regime.

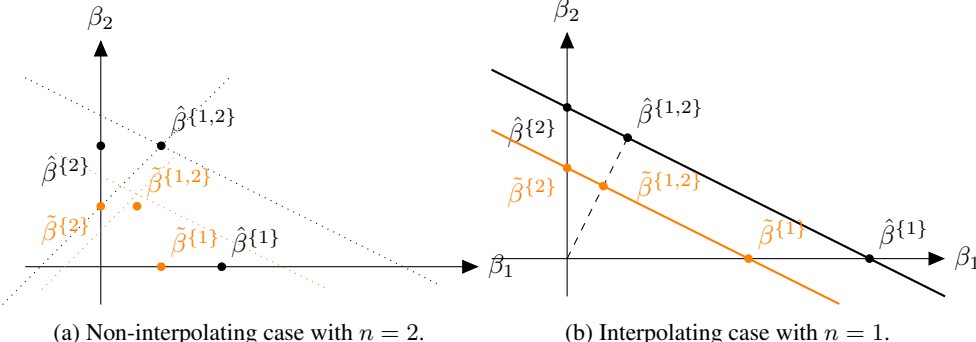

(a) Non-interpolating case with $n = 2$.      (b) Interpolating case with $n = 1$.

Figure 3: Minimal-norm least-squares solutions for draws $Y_A$ (black) and $Y_B$ (orange) for linear regression with $J = \{1, 2\}$, where $n$ varies

dataset, we impute smoking rates for a target state for a varying number of control states. We then discuss theoretical properties of the synthetic control estimator, which we connect to its imputation quality in the following Section 4.

## 3.1 Setup and Estimator

We consider a panel of $N + 1$ units observed over $T$ time periods, $Y = (y_{it})_{i \in \{0,\dots,N\}, t \in \{1,\dots,T\}} \in \mathbb{R}^{(N+1) \times T}$, where $i = 0$ denotes the target unit. Our goal is to impute $y_{0t}$ for $t \in \{T+1, \dots, T+S\}$ given $y_{it}$ for $i \in \{1, \dots, N\}, t \in \{T+1, \dots, T+S\}$ by the synthetic-control estimator $\hat{y}_{0t} = \sum_{i=1}^{N} \hat{w}_i y_{it}$ with convex weights $\hat{w} \in \mathcal{W} = \{w \in [0,1]^N; \sum_{i=1}^{n} w_i = 1\}$. Specifically, for a subset $J \subseteq \{1, \dots, N\}$ of control units, we consider the synthetic control weights

$$\hat{w}^J = \arg\min_{w \in \widehat{\mathcal{W}}^J} \|w\| \qquad\qquad \widehat{\mathcal{W}}^J = \arg\min_{w \in \mathcal{W}; w_j = 0 \forall j \notin J} \sum_{t=1}^{T} \left(y_{0t} - \sum_{i=1}^{N} w_i y_{it}\right)^2. \qquad (1)$$

Here, we choose the (unique) norm-minimizing synthetic control weights whenever there is more than one empirical risk minimizer. We can also interpret this solution as the limit $\hat{w}^J$ of a ridge penalized synthetic control estimator $\hat{w}_\eta^J$,

$$\hat{w}^J = \lim_{\eta \to 0} \hat{w}_\eta^J \qquad\qquad \hat{w}_\eta^J = \arg\min_{w \in \mathcal{W}; w_j = 0 \forall j \notin J} \sum_{t=1}^{T} \left(y_{0t} - \sum_{i=1}^{N} w_i y_{it}\right)^2 + \eta \|w\|^2, \qquad (2)$$

where $\hat{w}_\eta^J$ puts a penalty on the Euclidean norm $\|w\|^2$ of the weights, multiplied by a factor $\eta > 0$. This form of the penalized synthetic-control estimator is also considered by Shen et al. (2022).

We note that, unlike in the linear-regression case, we can now end up with non-interpolating solutions even in the case of high model complexity (many control units). The reason is that the convexity restriction $\hat{w} \in \mathcal{W}$ allows for interpolation only if the target outcomes are in the convex hull of the control outcomes.

## 3.2 Empirical Motivation

To illustrate some properties of the synthetic control estimator, we impute California smoking rates in the Abadie et al. (2010) dataset. In that dataset, California experiences the introduction of smoking legislation in 1989, for which Abadie et al. (2010) provides a causal effect estimate by imputing counterfactual smoking rates for those years when the legislation is in effect. We instead consider only the time before the legislation is introduced, giving us access to observed control outcomes in all years, even for California. Specifically, we fit synthetic control models for California on $T = 3$ years of data (1984–1986), and evaluate their imputation performance on the following $S = 2$ years (1987–1988) in terms of mean-squared error.

When fitting synthetic control models, we vary how many of the $N = 20$ control states we include in the estimation process. Specifically, for a given complexity $\ell \in \{1, \dots, N\}$, we average out-of-time

root-mean squared error (RMSE) across all $\binom{N}{\ell}$ combinations of control states. We report results in Figure 4.

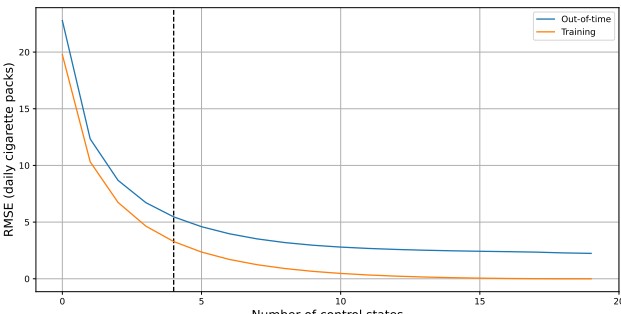

Figure 4: Average out-of-time (blue) and training (orange) RMSE for synthetic control for a varying number of control units.

Unlike the linear-regression case, the loss in the synthetic-control case changes monotonically: as we increase the number of control units, average RMSE decreases. We therefore observe a single-descent curve in the relationship of complexity and loss, with no notable change in regimes when the number of control states surpasses the number $T = 3$ of training periods.

As before, our illustration is extreme: by using only three training periods and a random selection of control states, we can provide a stark illustration of the difference in behavior between the synthetic control and linear-regression estimators.

### 3.3 Theoretical Properties and Graphical Illustration

While linear regression and synthetic control behave differently in terms of their double- vs single-descent behavior, we note that both exhibit continuing returns to increasing complexity, with no limit. In our empirical illustration, that return to complexity kicks in in the interpolation regime for linear regression and throughout for synthetic control. We now connect this commonality in returns to complexity to a corresponding theoretical connection.

**Proposition 4** (Model averaging for synthetic control). *For all $J$ with $|J| > 1$ and data $Y$ there exists*

$$\hat{\lambda} \in [0,1]^J, \sum_{j \in J} \hat{\lambda}_j = 1 \qquad such\ that \qquad \hat{w}^J = \sum_{j \in J} \hat{\lambda}_j \hat{w}^{J \setminus \{j\}}.$$

Hence, synthetic control has the same model-averaging property as interpolating linear regression (Proposition 1). Now, however, the model-averaging property also holds without interpolation, and is instead driven by the convexity of synthetic control weights. Furthermore, the weights can depend on outcome data, although only for pre-treatment outcomes. We further note that the result extends to penalized synthetic control with some fixed penalty parameter $\eta$.

We illustrate the model-averaging property for synthetic control in Figure 5, where we show that synthetic California with $|J| = 2$ (left) and $|J| = 3$ (right) control states is a convex combination of synthetic California with fewer control states. Here, we focus on the fit in the training data, and do not explicitly show the underlying synthetic-control weights. We note that Figure 5 covers both cases where the synthetic estimator for California has perfect training fit (right) and cases where it does not (left).

Unlike in the case of linear regression (Proposition 3) we do not, however, obtain an immediate bound on the variation of synthetic control models. Indeed, as the case of $|J| = 2$ controls in Figure 5 clarifies, the complex model can have more variance in weights than the simple ones (which here do not vary at all), despite the model-averaging property. In the next section, we will therefore connect model averaging directly to improvements of imputation quality, without relying on explicit variation results.

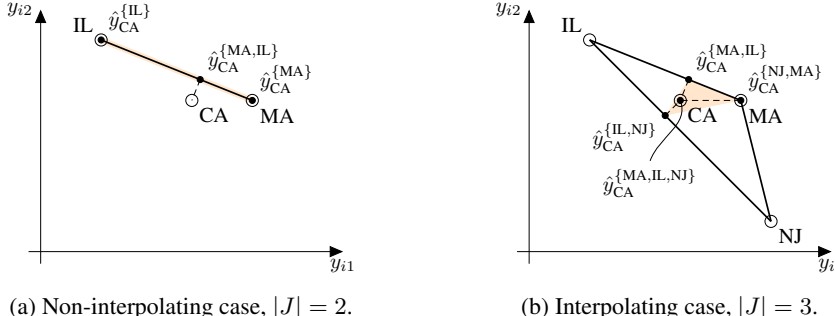

(a) Non-interpolating case, $|J| = 2$.
(b) Interpolating case, $|J| = 3$.

Figure 5: Synthetic-control examples for $T = 2$, where the set $J$ of included units varies.

## 4  Model-Averaging Based Risk Bounds

Above, we argued that more complex models are model averages over simpler models in two cases that are relevant to causal inference: interpolating linear regression and synthetic control. In this section, we discuss conditions under which model averaging leads to better imputation quality.

To unify the above cases, we now consider generic estimated functions $\hat{f}^* : \mathcal{X} \to \mathbb{R}$ that can be related to simpler models $\hat{f}^j : \mathcal{X} \to \mathbb{R}$ for an index set $j \in J$ by the model-averaging property

$$\hat{f}^* = \sum_{j \in J} \hat{\lambda}_j \hat{f}^j \qquad \text{for some} \qquad \hat{\lambda} \in [0,1]^J, \sum_{j \in J} \hat{\lambda}_j = 1. \tag{MA}$$

We see this model-averaging property as a purely mechanical property of estimators, which we applies to interpolating linear regression (Proposition 1) and to synthetic control (Proposition 4) by our results above

Model averaging provides some insurance against excess loss. Intuitively, being able to represent a more complex model $\hat{f}^*$ in terms of a model average over simpler models diversifies the risk of a bad imputation fit. When considering convex loss functions, we can make this intuition precise via a simple application of Jensen's inequality, which yields for the case of squared error that

$$(y - \hat{f}^*(x))^2 \leq \sum_{j \in J} \hat{\lambda}_j (y - \hat{f}^j(x))^2. \tag{3}$$

for any target point $(y, x) \in \mathbb{R} \times \mathcal{X}$. Hence, imputation loss using the complex model is at most a weighted average over the loss of simpler models. The relationship also extends directly to estimating averages of outcomes $y$ by averages of predictions $\hat{f}(x)$, as in the case of average treatment effects in Section 2.

In principle, the simple portfolio bound in (3) leaves open the possibility that the more complex model $\hat{f}^*$ performs as poorly as the worst of the simpler models $\hat{f}^j$. However, for this to occur, the weights would have to be positively correlated with *bad* performance. A condition on imputation quality we can therefore consider imposing is that the selection of weights is not, on average, working *against* imputation quality. The following result formalizes this idea on a (very) high level.

**Proposition 5** (Model-agnostic risk bound)**.** *Assume that* (MA) *holds and that for some distribution over training and target data we have that for all permutations* $\pi : J \to J$

$$\mathrm{E}[(y - \hat{f}^*(x))^2] \leq \mathrm{E}[(y - \hat{f}^*_\pi(x))^2] \qquad \text{where} \qquad \hat{f}^*_\pi = \sum_{j \in J} \hat{\lambda}_{\pi(j)} \hat{f}^j. \tag{P}$$

*Then we obtain the bound* $\mathrm{E}[(y - \hat{f}^*(x))^2] \leq \frac{1}{|J|} \sum_{j \in J} \mathrm{E}[(y - \hat{f}^j(x))^2].$

In words, if the model chosen by the data on average over some distribution is not worse than a model where we mix up weights, then the imputation performance of the complex model is not worse than the average of the imputation performances of simple models, leading to observations like those in Figures 1b and 4 where increased model complexity leads to improved imputation quality for randomly-ordered models.

We make two comments on the formal conditions of the proposition. First, it is enough to assume that (P) holds on average over permutations chosen uniformly at random. Second, the distribution behind the expectation E can incorporate prior distributions over underlying parameters, in which case the resulting bound holds on average over the same distribution.

In the examples of linear regression and synthetic control, Figure 6 illustrates the respective permuted models. In those cases, the permutation assumption (P) assumes that, on average, the model chosen by the data is not worse than a model where we mix up the weights (in gray).

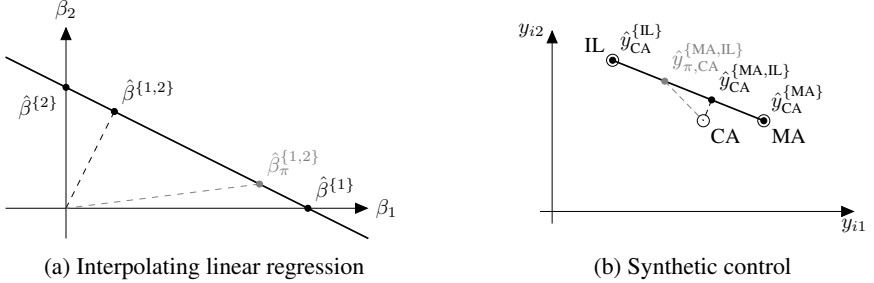

(a) Interpolating linear regression  (b) Synthetic control

Figure 6: Illustration of permutation bound based on the examples from Figures 2 and 5.

While more primitive conditions may be helpful to judge when a condition like (P) holds, we note two attractive properties. First, the assumption complements the model-averaging property (MA) in an important way: While model averaging relates more complex to less complex models, the permutation property only compares models of comparable complexity (assuming that there are no systematic ex-ante differences between the $\hat{f}^j$). To violate this property would thus amount to assuming that selection among models with *comparable* complexity is disadvantageous, which may be unreasonable to expect on average. Second, we can formulate this condition on the level of estimators, without explicit reference to the underlying data-generating process.

## 5  Conclusion

We study the imputation performance of interpolating linear regression and synthetic control, and provide a unified perspective on returns to complexity in both cases: More complex models can be expressed as model averages over simpler ones. While we provide some high-level assumptions on when this model-averaging property improves average imputation risk, more work is needed to establish primitive sufficient conditions. This includes, in particular, studying how the bias changes as models become more complex. In addition, we limit our analysis to comparing more complex to simpler models when features or control units are randomly ordered, but in practice, we may have knowledge about which simple models are more plausible than others. However, our results show that highly over-parameterized models that achieve perfect in-sample fit can yield measurable performance improvements over non-random simple models in causal settings. Future research could explore conditions under which this phenomenon holds, namely where complex models can beat non-random simple ones.

## Acknowledgments and Disclosure of Funding

We thank Tengyuan Liang, Sendhil Mullainathan, Ashesh Rambachan, audiences at Emory University, the 2023 "Econometrics in the Era of Machine Learning" conference at the University of Chicago, the 2023 "HDMetrics: Big Data, High-Dimensional Methods, and Machine Learning" workshop at the University of Illinois at Urbana-Champaign, the 2023 CEME Conference for Young Econometricians at Georgetown University, and the 2023 California Econometrics Conference at the University of Washington, as well as four anonymous reviewers for helpful comments and discussions.

Computational support was provided by the Data, Analytics, and Research Computing (DARC) group at the Stanford Graduate School of Business (RRID:SCR_022938).

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

# A Proofs

*Proof of Proposition 1.* We provide a direct proof for the choice $\lambda_j = \frac{1-X_j'(X_JX_J')^{-1}X_j}{|J|-n}$ via the Sherman–Morrison–Woodbury formula. We first note that, for $k \geq n$, $A \in \mathbb{R}^{n\times k}$ of full row rank $n$, $a \in \mathbb{R}^n$, and $\hat{\alpha} = \arg\min_{\alpha \in \mathbb{R}^k; A\alpha=a} \|\alpha\|$ we have that $\hat{\alpha} = A'(AA')^{-1}a$. Indeed, $A\hat{\alpha} = a$, and for any $\alpha \in \mathbb{R}^k$ with $A\alpha = a$ and $\alpha \neq \hat{\alpha}$, for $\Pi = A'(AA')^{-1}A$ we have that

$$\|\alpha\|^2 = \|\Pi\alpha\|^2 + \|(\mathbb{I}-\Pi)\alpha\|^2 = \|\Pi\hat{\alpha}\|^2 + \|(\mathbb{I}-\Pi)(\alpha-\hat{\alpha})\|^2 = \|\hat{\alpha}\|^2 + \|\alpha-\hat{\alpha}\|^2 > \|\hat{\alpha}\|^2.$$

We next write $X^J \in \mathbb{R}^{n\times k}$ for the matrix with columns $X_j^J = X_j$ for $j \in J$ and $X_j^J = \mathbf{0}$ for $j \notin J$. Applying the above result to $\hat{\beta}^J$ and $\hat{\beta}^{J\setminus\{j\}}$ for all $j \in J$, we find

$$\hat{\beta}^J = X^{J\prime}(X_JX_J')^{-1}Y, \qquad\qquad \hat{\beta}^{J\setminus\{j\}} = X^{J\setminus\{j\}\prime}(X_{J\setminus\{j\}}X_{J\setminus\{j\}}')^{-1}Y.$$

Using that $X_{J\setminus\{j\}}X_{J\setminus\{j\}}' = X_JX_J' - X_jX_j'$, which is invertible by the assumption that $X_{J\setminus\{j\}}$ is of full row rank, we find by the Sherman–Morrison–Woodbury that

$$(X_{J\setminus\{j\}}X_{J\setminus\{j\}}')^{-1} = (X_JX_J')^{-1} + (X_JX_J')^{-1}X_j(1 - X_j'(X_JX_J')^{-1}X_j)^{-1}X_j'(X_JX_J')^{-1} \quad (4)$$

with $X_j'(X_JX_J')^{-1}X_j \neq 1$. Plugging in,

$$\hat{\beta}^{J\setminus\{j\}} = (X^J - X^{\{j\}})'(X_{J\setminus\{j\}}X_{J\setminus\{j\}}')^{-1}Y$$

$$= \hat{\beta}^J - X^{\{j\}\prime}(X_JX_J')^{-1}Y - X^{\{j\}\prime}(X_JX_J')^{-1}Y\frac{X_j'(X_JX_J')^{-1}X_j}{1 - X_j'(X_JX_J')^{-1}X_j}$$

$$+ X^{J\prime}(X_JX_J')^{-1}X_jX_j'(X_JX_J')^{-1}Y\frac{1}{1 - X_j'(X_JX_J')^{-1}X_j}$$

$$= \hat{\beta}^J + \left(X^{J\prime}(X_JX_J')^{-1}X_jX_j'(X_JX_J')^{-1} - X^{\{j\}\prime}(X_JX_J')^{-1}\right)Y\frac{1}{1 - X_j'(X_JX_J')^{-1}X_j}.$$

Since $\sum_{j\in J} X_jX_j' = X_JX_J'$ and $\sum_{j\in J} X^{\{j\}} = X^J$, we have that

$$\sum_{j\in J} \hat{\beta}^{J\setminus\{j\}}\lambda_j = \hat{\beta}^J\sum_{j\in J}\lambda_j + (|J|-n)(X^{J\prime}(X_JX_J')^{-1}X_JX_J'(X_JX_J')^{-1}Y - X^{J\prime}(X_JX_J')^{-1}Y) = \hat{\beta}^J\sum_{j\in J}\lambda_j.$$

Finally, $X_j'(X_JX_J')^{-1}X_j \geq 0$ since $X_JX_J'$ positive definite, $X_j'(X_JX_J')^{-1}X_j \leq 1$ since $X_{J\setminus\{j\}}X_{J\setminus\{j\}}' = X_JX_J' - X_jX_j' \preceq X_JX_J'$ in (4), and $\sum_{j\in J} X_j'(X_JX_J')^{-1}X_j = \mathrm{tr}\left(\sum_{j\in J}X_jX_j'X_JX_J'\right) = n$, so $\lambda_j \in [0,1]$ for all $j \in J$ and $\sum_{j=1}^J \lambda_j = 1$. $\qquad\square$

*Proof of Proposition 2.* The result follows from Proposition 3 by noting that, for two indpendent draws $Y_A$ and $Y_B$ for fixed $X$, we have that

$$\mathrm{E}[\|\hat{\beta}_A^J - \hat{\beta}_B^J\|^2|X] = \mathrm{E}[\|(\hat{\beta}_A^J - \mathrm{E}[\hat{\beta}^J|X]) - (\hat{\beta}_B^J - \mathrm{E}[\hat{\beta}^J|X])\|^2|X]$$

$$= \mathrm{E}[\|\hat{\beta}_A^J - \mathrm{E}[\hat{\beta}^J|X]\|^2|X] + \mathrm{E}[\|\hat{\beta}_B^J - \mathrm{E}[\hat{\beta}^J|X]\|^2|X] = 2\,\mathrm{tr}\,\mathrm{Var}(\hat{\beta}^J|X)$$

(and the same for $J\setminus\{j\}$), and thus

$$\mathrm{tr}\,\mathrm{Var}(\hat{\beta}^J|X) = \frac{1}{2}\,\mathrm{E}[\|\hat{\beta}_A^J - \hat{\beta}_B^J\|^2|X] \leq \frac{1}{2}\,\mathrm{E}\left[\min_{j\in J}\|\hat{\beta}_A^{J\setminus\{j\}} - \hat{\beta}_B^{J\setminus\{j\}}\|^2\bigg|X\right]$$

$$\leq \min_j \frac{1}{2}\mathrm{E}[\|\hat{\beta}_A^{J\setminus\{j\}} - \hat{\beta}_B^{J\setminus\{j\}}\|^2|X] = \mathrm{tr}\,\mathrm{Var}(\hat{\beta}^{J\setminus\{j\}}|X). \qquad\square$$

*Proof of Proposition 3.* Consider first the case $|J| \leq n$. Under Assumption 1, we define the projection matrices

$$\Pi^J = X_J(X_J'X_J)^{-1}X_J' \in \mathbb{R}^{n\times n}, \quad \Pi^{J\setminus\{j\}} = X_{J\setminus\{j\}}(X_{J\setminus\{j\}}'X_{J\setminus\{j\}})^{-1}X_{J\setminus\{j\}}' \in \mathbb{R}^{n\times n}.$$

Since $\Pi^{J\setminus\{j\}} = \Pi^{J\setminus\{j\}}\Pi^J$, we have that $X\hat{\beta}^{J\setminus\{j\}} = \Pi^{J\setminus\{j\}}Y = \Pi^{J\setminus\{j\}}\Pi^J Y = \Pi^{J\setminus\{j\}}\hat{\beta}^J$. Therefore,

$$
\begin{aligned}
\|\hat{\beta}_A^J - \hat{\beta}_B^J\|_{X'X}^2 &= \|X\hat{\beta}_A^J - X\hat{\beta}_B^J\|^2 = \|\Pi^{J\setminus\{j\}}(X\hat{\beta}_A^J - X\hat{\beta}_B^J)\|^2 + \|(\mathbb{I} - \Pi^{J\setminus\{j\}})(X\hat{\beta}_A^J - X\hat{\beta}_B^J)\|^2 \\
&\geq \|\Pi^{J\setminus\{j\}}(X\hat{\beta}_A^J - X\hat{\beta}_B^J)\|^2 = \|X\hat{\beta}_A^{J\setminus\{j\}} - X\hat{\beta}_B^{J\setminus\{j\}}\|^2 = \|\hat{\beta}_A^{J\setminus\{j\}} - \hat{\beta}_B^{J\setminus\{j\}}\|_{X'X}^2.
\end{aligned}
$$

Consider now the case $|J| > n$. Using the notation from the proof of Proposition 1, under Assumption 1 we have that $X^J\hat{\beta}^J = X\hat{\beta}^J = Y = X\hat{\beta}^{J\setminus\{j\}} = X^J\hat{\beta}^{J\setminus\{j\}}$ and thus $\Pi\hat{\beta}^J = \Pi\hat{\beta}^{J\setminus\{j\}}$ (as well as $(\mathbb{I} - \Pi)\hat{\beta}^J = \mathbf{0}$) for the projection matrix $\Pi = X^{J\prime}(X_J X'_J)^{-1}X^J \in \mathbb{R}^{k\times k}$. As a consequence,

$$
\begin{aligned}
\|\hat{\beta}_A^J - \hat{\beta}_B^J\|^2 &= \|\Pi(\hat{\beta}_A^J - \hat{\beta}_B^J)\|^2 + \|(\mathbb{I} - \Pi)(\hat{\beta}_A^J - \hat{\beta}_B^J)\|^2 = \|\Pi(\hat{\beta}_A^{J\setminus\{j\}} - \hat{\beta}_B^{J\setminus\{j\}})\|^2 \\
&\leq \|\Pi(\hat{\beta}_A^{J\setminus\{j\}} - \hat{\beta}_B^{J\setminus\{j\}})\|^2 + \|(\mathbb{I} - \Pi)(\hat{\beta}_A^{J\setminus\{j\}} - \hat{\beta}_B^{J\setminus\{j\}})\|^2 = \|\hat{\beta}_A^{J\setminus\{j\}} - \hat{\beta}_B^{J\setminus\{j\}}\|^2. \quad\square
\end{aligned}
$$

*Proof of Proposition 4.* Building upon the notation from Section 3.1, for $J \subset \{1,\ldots,N\}$ write $\mathcal{W}^J = \{w \in \mathcal{W}; w_j = 0 \text{ for all } j \notin J\}$ (where $\mathcal{W} = \{w \in [0,1]^N; \sum_{i=1}^N w_i = 1\}$ is the $N-1$-simplex) and let $\partial\mathcal{W}^J = \bigcup_{j\in J}\mathcal{W}^{J\setminus\{j\}} \subseteq \mathcal{W}^J$ be the boundary of $\mathcal{W}^J$. For outcomes, it will also be convenient to write $X = (y_{it})_{t\in\{1,\ldots,T\},i\in\{1,\ldots,N\}} \in \mathbb{R}^{T\times N}$ for the pre-treatment outcomes of the control units (with columns representing units), and $y = (y_{0t})_{t\in\{1,\ldots,T\}} \in \mathbb{R}^T$ for the pre-treatment outcomes of the treated unit.

As the first step, we note that we can express the quality of synthetic control weights $w \in \mathcal{W}^J$ as

$$
\|Xw - y\|^2 = \|Xw - \overline{y}^J\|^2 + \|\overline{y}^J - y\|^2 \tag{5}
$$

in terms of the fitted values $\overline{y}^J = X\overline{w}^J$ for the solution $\overline{w}^J$ to a relaxed problem that drops the non-negativity constraint. That solution with weights in $\mathcal{W}^* = \{w \in \mathbb{R}^N; \sum_{i=1}^n w_i = 1\}$ is defined, analogously to the synthetic-control solution in (1), as

$$
\overline{w}^J = \arg\min_{w\in\overline{\mathcal{W}}^J}\|w\| \in \mathcal{W}^*, \qquad \overline{\mathcal{W}}^J = \arg\min_{w\in\mathcal{W}^*; w_j=0\forall j\notin J}\|Xw - y\| \subseteq \mathcal{W}^*.
$$

For this solution, we note that $\|Xw - y\|^2 = \|X(w - \overline{w}^J)\|^2 + 2(w - \overline{w}^J)'X'(X\overline{w}^J - y) + \|X\overline{w}^J - y\|^2$. Assume now that $(w - \overline{w}^J)'X'(X\overline{w}^J - y) \neq 0$. Then there is some $\varepsilon \neq 0$ such that $\overline{w}^J(\varepsilon) = \overline{w}^J - (w - \overline{w}^J)\varepsilon \in \mathcal{W}^*$ with $w_j = 0$ for $j \notin J$ fulfills $\|X\overline{w}^J(\varepsilon) - y\| < \|X\overline{w}^J - y\|$, contradicting the choice of $\overline{w}^J$. Hence we must have that $\|Xw - y\|^2 = \|X(w - \overline{w}^J)\|^2 + \|X\overline{w}^J - y\|^2 = \|Xw - \overline{y}^J\|^2 + \|\overline{y}^J - y\|^2$, which establishes (5).

As the second step, we note that we can therefore define the synthetic control solution in (1) in terms of the fitted values $\overline{y}^J$ of the relaxed solution as

$$
\hat{w}^J = \arg\min_{w\in\widehat{\mathcal{W}}^J}\|w\| \in \mathcal{W}^J, \qquad \widehat{\mathcal{W}}^J = \arg\min_{w\in\mathcal{W}^J}\|Xw - \overline{y}^J\| \subseteq \mathcal{W}^J.
$$

This follows immediately from (5) since $\|\overline{y}^J - y\|$ is not affected by the choice of $w \in \mathcal{W}^J$. Similarly, for the constrained solutions with index set $J \setminus \{j\}$ for $j \in J$, we have that

$$
\hat{w}^{J\setminus\{j\}} = \arg\min_{w\in\widehat{\mathcal{W}}^{J\setminus\{j\}}}\|w\| \in \mathcal{W}^{J\setminus\{j\}}, \qquad \widehat{\mathcal{W}}^{J\setminus\{j\}} = \arg\min_{w\in\mathcal{W}^{J\setminus\{j\}}}\|Xw - \overline{y}^J\| \subseteq \mathcal{W}^{J\setminus\{j\}}
$$

since $\mathcal{W}^{J\setminus\{j\}} \subseteq \mathcal{W}^J$ for all $j \in J$.

As the third (and central) step, we use Farkas' lemma to argue that there exist $\lambda \in \mathbb{R}^J$ with $\lambda_j \geq 0$ for all $j \in J$ such that $X\overline{w}^J = \sum_{j\in J}\lambda_j X\overline{w}^{J\setminus\{j\}}$.

Assume first that $X\hat{w}^J \neq \overline{y}^J$. Then we must have that $\hat{w}^J \in \partial\mathcal{W}^J$. Indeed, if $\hat{w}^J \in \mathcal{W}^J \setminus \partial\mathcal{W}^J$ then there exists some $\varepsilon > 0$ such that $\hat{w}^J(\varepsilon) = \hat{w}^J(1 - \varepsilon) + \overline{w}^J\varepsilon \in \mathcal{W}^J$, for which $\|X\hat{w}^J(\varepsilon) - \overline{y}^J\| = \|X(\hat{w}^J(\varepsilon) - \overline{w}^J)\| = (1 - \varepsilon)\|X(\hat{w}^J - \overline{w}^J)\| < \|X\hat{w}^J - \overline{y}^J\|$, contradicting the choice of $\widehat{\mathcal{W}}^J$ and

$\hat{w}^J$. Hence $\hat{w}^J \in \partial\mathcal{W}^J$, so there is some $j$ with $\hat{w}^J \in \mathcal{W}^{J\setminus\{j\}}$, which implies that $\hat{w}^{J\setminus\{j\}} = \hat{w}^J$ and $X\hat{w}^J = X\hat{w}^{J\setminus\{j\}}$. This means that we can choose $\lambda$ as the indicator for component $j$.

Assume now that $X\hat{w}^J = \overline{y}^J$, and that there exists no such $\lambda$. Then, by Farkas' lemma, there exists $v \in \mathbb{R}^T \setminus \{\mathbf{0}\}$ such that $v'X\hat{w}^J < 0$ and $v'X\hat{w}^{J\setminus\{j\}} \geq 0$ for all $j \in J$. Define the projection matrix $\Pi = \frac{vv'}{v'v} \in \mathbb{R}^{T \times T}$, and let $\mathcal{W}^* = \arg\min_{w \in \mathcal{W}^J; \Pi X(w-\hat{w}^J) = X(w-\hat{w}^J)} v'Xw \subseteq \mathcal{W}^J$. Then the minimum is attained at a boundary point $w^* \in \mathcal{W}^* \cap \partial\mathcal{W}^J$ of the feasible set. Indeed, the feasible set is non-empty since it includes $\hat{w}^J$, and it is compact and convex. The minimum of the linear function is therefore attained at a boundary point, which is in $\partial\mathcal{W}^J$. As a consequence, $w^* \in \mathcal{W}^{J\setminus\{j\}}$ for some $j \in J$. Since $\hat{w}^J \in \mathcal{W}^J$, we have that $v'Xw^* \leq v'X\hat{w}^J < v'X\hat{w}^{J\setminus\{j\}}$. Hence there is some $\varepsilon \in (0,1]$ such that $\hat{w}^{J\setminus\{j\}}(\varepsilon) = \hat{w}^{J\setminus\{j\}}(1-\varepsilon) + w^*\varepsilon \in \mathcal{W}^{J\setminus\{j\}}$ fulfills $v'X\hat{w}^{J\setminus\{j\}}(\varepsilon) = v'X\hat{w}^J$. Since we therefore have $\Pi X\hat{w}^{J\setminus\{j\}}(\varepsilon) = \Pi X\hat{w}^J$, as well as $(\mathbb{I} - \Pi)X\hat{w}^{J\setminus\{j\}}(\varepsilon) = (\mathbb{I} - \Pi)X(\hat{w}^{J\setminus\{j\}}(1-\varepsilon) + \varepsilon\hat{w}^J)$ since $\Pi X(w^* - \hat{w}^J) = X(w^* - \hat{w}^J)$, we have that

$$\|X\hat{w}^{J\setminus\{j\}}(\varepsilon) - \overline{y}^J\|^2 = \|X(\hat{w}^{J\setminus\{j\}}(\varepsilon) - \hat{w}^J)\|^2$$
$$= \|\Pi X(\hat{w}^{J\setminus\{j\}}(\varepsilon) - \hat{w}^J)\|^2 + \|(\mathbb{I}-\Pi)X(\hat{w}^{J\setminus\{j\}}(\varepsilon) - \hat{w}^J)\|^2 = 0 + (1-\varepsilon)^2\|(\mathbb{I}-\Pi)X(\hat{w}^{J\setminus\{j\}} - \hat{w}^J)\|^2$$
$$< \|\Pi X(\hat{w}^{J\setminus\{j\}} - \hat{w}^J)\|^2 + \|(\mathbb{I}-\Pi)X(\hat{w}^{J\setminus\{j\}} - \hat{w}^J)\|^2 = \|X(\hat{w}^{J\setminus\{j\}} - \hat{w}^J)\|^2 = \|X\hat{w}^{J\setminus\{j\}} - \overline{y}^J\|^2,$$

contradicting the choice of $\hat{w}^{J\setminus\{j\}}$. Hence, such $\lambda$ must exist.

As the fourth step, we expand the previous result on fitted values to the weights themselves in the case of penalized synthetic control, and show that the weights sum to one in that case. To this end, note that we can write the penalized synthetic control estimator from (2) as $\hat{w}_\eta^J = \arg\min_{w \in \mathcal{W}^J} \|Xw - y\|^2 + \eta\|w\|^2$. Write now $\tilde{X}_\eta^J = (X_J'X_J + \eta\mathbb{I})^{1/2} \in \mathbb{R}^{J \times J}$ for the symmetric positive-definite matrix square root of the symmetric positive-definite $X_J'X_J + \eta\mathbb{I}$, where $X_J$ is a matrix of the columns of $X$ with index in $J$, and $\tilde{y}_\eta^J = (\tilde{X}_\eta^J)^{-1}X_J'y \in \mathbb{R}^J$. For $w_J$ the entries of $w \in \mathcal{W}^J$ corresponding to the index set $J$, we find

$$\|Xw - y\|^2 + \eta\|w\|^2 = \|X_J w_J - y\|^2 + \eta\|w_J\|^2$$
$$= w_J'X_J'X_J w_J - 2w_J'X_J'y + y'y + \eta w_J'w_J = w_J'(X_J'X_J + \eta\mathbb{I})w_J - 2w_J'X_J'y + y'y$$
$$= w_J'\tilde{X}_\eta^{J'}\tilde{X}_\eta^J w_J - 2w_J'\tilde{X}_\eta^{J'}\left((\tilde{X}_\eta^J)^{-1}X_J'y\right) + y'y = \|\tilde{X}_\eta^J w_J - \tilde{y}_\eta^J\|^2 - \|\tilde{y}_\eta^J\|^2 + \|y\|^2.$$

Hence, we can write (noting that $\mathcal{W}^{J\setminus\{j\}} \subseteq \mathcal{W}^J$)

$$\hat{w}_\eta^J = \arg\min_{w \in \mathcal{W}^J} \|\tilde{X}_\eta^J w_J - \tilde{y}_\eta^J\|, \qquad w_\eta^{J\setminus\{j\}} = \arg\min_{w \in \mathcal{W}^{J\setminus\{j\}}} \|\tilde{X}_\eta^J w_J - \tilde{y}_\eta^J\|,$$

so we can interpret penalized synthetic control on units $J$ and $J \setminus \{j\}$ with time periods $\{1, \ldots, T\}$ and the original outcomes as non-penalized synthetic control on units $J$ and $J \setminus \{j\}$ with time periods $J$ and transformed outcomes, where we note that the synthetic-control solutions are unique in this case. Hence, we can apply the previous result to conclude that there exists $\lambda_\eta \in \mathbb{R}^J$ with $\lambda_{\eta,j} \geq 0$ for all $j \in J$ such that $\tilde{X}_\eta^J\tilde{w}_\eta^J = \sum_{j \in J}\lambda_{\eta,j}\tilde{X}_\eta^J\tilde{w}_\eta^{J\setminus\{j\}}$. Since $\tilde{X}_\eta^J$ is invertible, it now also follows that $\tilde{w}_\eta^J = \sum_{j \in J}\lambda_{\eta,j}\tilde{w}_\eta^{J\setminus\{j\}}$. Since also $\tilde{w}_\eta^J \in \mathcal{W}$ and $\tilde{w}_\eta^{J\setminus\{j\}} \in \mathcal{W}$ for all $j \in J$, we have that $\sum_{j \in J}\lambda_{\eta,j} = \sum_{j \in J}\lambda_{\eta,j}\mathbf{1}'\tilde{w}_\eta^{J\setminus\{j\}} = \mathbf{1}'\tilde{w}_\eta^J = 1$. This establishes the main claim of the proposition for penalized synthetic control.

As the fifth and final step, we derive the main result on minimum-norm synthetic control from the above results on penalized synthetic control. Consider some sequence $(\eta_\iota)_{\iota=1}^\infty$ in $(0,\infty)$ with $\eta_\iota \to 0$, and for every $\iota$ apply the previous step to the penalized synthetic control estimator with penalty $\eta_\iota$ to obtain a weight vector $\lambda_{\eta_\iota} \in \Lambda^J = \{\lambda \in [0,1]^J; \sum_{j=1}^J \lambda_j = 1\}$. Since $\Lambda^J$ is compact, $(\lambda_{\eta_\iota})_{\iota=1}^\infty$ must have a converging subsequence with some limit $\lambda \in \Lambda^J$. Using the limit along this subsequence, we have that

$$\hat{w}^J = \lim_{\iota \to \infty}\hat{w}_{\eta_\iota}^J = \lim_{\iota \to \infty}\sum_{j \in J}\lambda_{\eta_\iota,j}\tilde{w}_{\eta_\iota}^{J\setminus\{j\}} = \sum_{j \in J}\left(\lim_{\iota \to \infty}\lambda_{\eta_\iota,j}\right)\left(\lim_{\iota \to \infty}\tilde{w}_{\eta_\iota}^{J\setminus\{j\}}\right) = \sum_{j \in J}\lambda_j\tilde{w}^{J\setminus\{j\}}. \quad \square$$

*Proof of Proposition 5.* By Jensen's inequality applied to an average over the bounds in (3),

$$\mathrm{E}[(y - \hat{f}^*(x))^2] \leq \frac{1}{|J|!} \sum_{\pi} \mathrm{E}[(y - \hat{f}^*_\pi(x))^2] \leq \sum_{j \in J} \mathrm{E}\left[ \underbrace{\frac{1}{|J|!} \sum_{\pi} \hat{\lambda}_{\pi(j)}}_{= \frac{1}{|J|}} (y - \hat{f}^j(x))^2 \right]. \qquad \square$$

## B  Details of the Empirical Illustrations

### B.1  Many-Regressor Linear Least-Squares on CPS Data

We utilize the publicly available[5] CPS control and NSW experimental control datasets, drawn from the study presented in LaLonde (1986) as used by Dehejia and Wahba (1999, 2002). The resulting data has 15,992 observations for CPS and 260 for NSW, with both datasets containing an identical set of variables, detailed in Table 1.

| Variable | Data Type | Description |
|---|---|---|
| age | Discrete | Age |
| education | Discrete | Years of education |
| black | Dummy | Black |
| hispanic | Dummy | Hispanic |
| married | Dummy | Marital status |
| nodegree | Dummy | Lack of college degree |
| re74 | Continuous | Income in 1974 |
| re75 | Continuous | Income in 1975 |
| re78 | Continuous | Income in 1978 |

Table 1: CPS and NSW dataset variables

We use `re78` as the outcome variable and all other variables as covariates. In order to achieve high dimensionality, we first discretize the continuous income covariates into 50 bins via quantile binning. We then construct a series of dummies for each discrete variable, corresponding to indicators for each discretized value. We then interact all these dummy variables, as well as those covariates which were originally dummies, taking care not to interact those which are mutually exclusive (e.g. originating from the same original covariate or corresponding to race). We then drop any interactions that are zero for all observations in the data. The resulting transformed dataset contains 8,408 dummy covariates, as well as the unmodified outcome variable. In order to ensure that the covariate matrix is full row rank for an arbitrary subset of columns, we go on to add iid $\mathcal{N}(0, 0.0004)$ noise to each of the covariate values (again leaving the outcome variable unaffected). We then select a random subset of 3,000 observations from the CPS dataset as our in-sample set, using the 260 NSW observations as our out-of-sample set.

For fitting models of varying complexity, we randomly permute the order of the columns of the covariate matrix; denote the resulting matrix as $X$. We then add an intercept and iterate over varying levels of complexity $\ell$, ranging from 1 to 8,409, corresponding to the number of covariates that we will use for estimation. We then estimate the OLS coefficient vector $\hat{\beta}^\ell = X^{\ell\dagger}y$, where $\dagger$ denotes the Moore–Penrose pseudoinverse.

To evaluate performance, we first select a sample size $m$ and then draw 1000 samples of size $m$ from the NSW set. We then take define our evaluation metric to be the RMSE across these 1000 samples, with error defined as the difference between the average predicted outcome and the true average outcome for each sample:

$$\mathrm{RMSE}(\ell, m) = \sqrt{\frac{1}{1000} \sum_{j=1}^{1000} \left[ \left( \frac{1}{m} \sum_{i=1}^{m} y^*_{ji} \right) - \left( \frac{1}{m} \sum_{i=1}^{m} x^{*\top}_{ji} \hat{\beta}^\ell \right) \right]^2}$$

---

[5] users.nber.org/~rdehejia/data/.nswdata2.html.  We use the files corresponding to cps_controls.txt and nswre74_control.txt.

This metric assesses the ability of the given model to accurately predict mean outcomes, a quantity of direct relevance to ATE estimaiton. Where $X^*, y^*$ denote the out-of-sample covariate and outcome variables, respectively, and can correspond to either the CPS or NSW held-out samples. The subscript $ji$ refers to the $i$th observation of the $j$th sample of size $m$. In order to smooth out the effects of the random ordering of columns, we repeat this exercise for five different random orderings and take a pointwise average to obtain smooth RMSE vs. complexity curves (Figure 1b). A zoomed-in version of this plot, focusing on the highly overparametrized regime and emphasizing the ability of some of the depicted models to outperform simple baselines can be found in Figure 7.

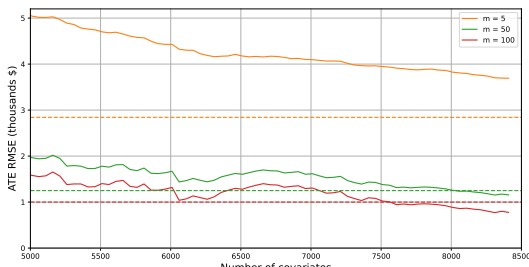

Figure 7: Average RMSE for linear regression for a varying number of covariates ($\ell > 5000$).

As a further illustration of the effects of increasing complexity, Figure 8 shows the average norm of the model coefficients across different model complexities, with the average taken over random covariate orderings as before. Starting small, model coefficients initially grow (in terms of their Euclidean norm), reaching their peak at the interpolation threshold. To the right of the interpolation threshold, the norm of the model coefficients decreases mechanically, since the estimator now minimizes the norm among all interpolating solutions with fewer and fewer sparsity constraints.

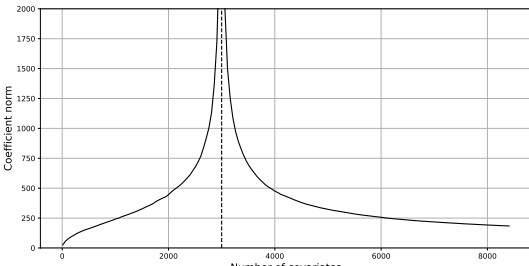

Figure 8: Average of the coefficient norm $\|\hat{\beta}^J\|$ for varying size of the set $J$ of covariates.

## B.2   Many-Unit Synthetic Control on Smoking Data

For our synthetic-control exercise, we utilize public data from the Centers for Disease Control and Prevention[6] containing annual cost, revenue, tax, and quantity data for cigarette sales by state for the years 1970 to 2019. We follow the approach of Abadie et al. (2010) in using synthetic control to estimate per-capita cigarette pack consumption for the target state, California, as a function of the other 49 states and Washington, D.C. For our evaluation, we utilize two years of data (1987 and 1988) as a hold-out sample and fit the model on three years (1984 to 1986). All of our data precedes the year in which anti-smoking legislation took effect in California (1989).

We begin by selecting a random subset of 20 states to serve as our donor pool, for computational tractability. We then select a level of complexity $\ell$ and select a subset of $\ell$ states from the chosen 20. Using that subset, we then estimate synthetic control weights based on the in-sample period,

---

[6]chronicdata.cdc.gov/Policy/The-Tax-Burden-on-Tobacco-1970-2019/7nwe-3aj9.

choosing convex weight vector $\hat{w}^{\ell} \in \mathcal{W} = \{w \in [0,1]^N; \sum_{i=1}^{n} w_i = 1\}$ as described in Section 3.1:

$$\hat{w}^{\ell} = \underset{w \in \widehat{\mathcal{W}}^{\ell}}{\arg\min} \|w\| \qquad \widehat{\mathcal{W}}^{\ell} = \underset{w \in \mathcal{W}; w_j = 0 \forall j \notin J}{\arg\min} \sum_{t=1984}^{1986} (y_{0t} - \sum_{i=1}^{\ell} w_i y_{it})^2.$$

Here, $y_0$ denotes the target state, California. We then compute the out-of-sample prediction error as:

$$\text{RMSE}(\ell) = \sqrt{\frac{1}{2} \sum_{t=1987}^{1988} (y_{0t} - \sum_{i=1}^{\ell} \hat{w}_i^{\ell} y_{it})^2}$$

We then iterate over all $\binom{20}{\ell}$ possible combinations of donor units for the given complexity level and take the average RMSE value to be the predictive error for the given complexity level. We vary $\ell$ from 1 to 20 to trace out the curve of synthetic control prediction risk vs. complexity (Figure 4).

As a robustness check, we also consider synthetic control with 10 pre-treatment periods, where the included control states are chosen among all 50 available donors (49 other states and Washington, D.C.). In particular, at each complexity level $\ell$ we consider $\min\{\binom{50}{\ell}, 10000\}$ combinations of donor units. The results are qualitatively similar, and provided in Figure 9.

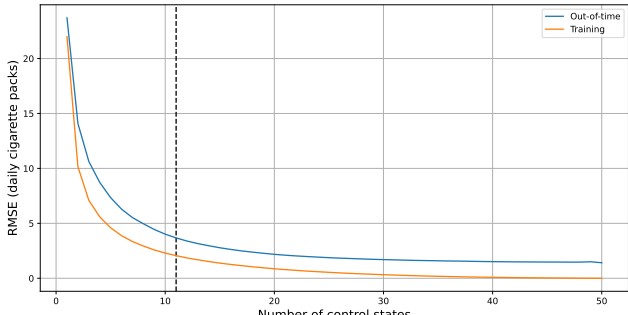

Figure 9: Average out-of-time (blue) and training (orange) RMSE for synthetic control for a varying number of control units as in Figure 4, but with ten pre-treatment periods and control units chosen randomly among all available donors.

