# OpenReview forum: "Double and Single Descent in Causal Inference with an Application to High-Dimensional Synthetic Control"
_NeurIPS.cc/2023/Conference — NeurIPS 2023 poster_

### Official Review · Reviewer_aRQr · 2023-07-05

**Soundness:** 3 good
**Presentation:** 3 good
**Contribution:** 2 fair
**Rating:** 6
**Confidence:** 3

**Summary:**

This paper investigates over-parametrized models in causal inference, specifically focusing on high-dimensional linear regression models and high-dimensional synthetic control estimators with a large number of control units. The authors examine the prediction risk behaviors associated with these two estimators, and present a unified theoretical perspective on the descent phenomena in the interpolating (=high-dimensional) regime, which they refer to as the “model-averaging property” as demonstrated in Proposition 1 and Proposition 4; also see Eq. (MA). By introducing the high-level assumption of the model-averaging property (MA), which posits that the full model can be expressed as a convex combination of simpler “leave-one-out” counterparts, and another technical assumption (P) concerning the optimality of the aggregated model against a random permutation of the weights, the authors assert that the complex model cannot perform worse than the average performance of the simpler models (Proposition 5).

In summary, this paper provides a concise exposition of the “benign overfitting” phenomena observed in two estimators widely utilized in causal inference. The authors rely on two high-level geometric assumptions, the model-averaging property and Assumption (P), which provide insights into the descent behavior of these estimators and warrant further investigations in follow-up studies.

**Strengths:**

This paper offers a simple yet intriguing perspective on understanding the double descent phenomenon in the interpolating regime by leveraging the mechanical properties of predictive estimators that are largely agnostic about the underlying data generating process. The authors adeptly combine knowledge from linear algebra and convex analysis to provide concise explanations for the returns of complex models in high-dimensional causal estimators. The practical implications of their findings, particularly in the context of synthetic control with many control units, are noteworthy. The paper alleviates concerns about overfitting when using a large number of control units, eliminating the need for pre-selecting an appropriate subset, which could be challenging in practice.

Furthermore, the authors' focused approach on linear regression models and synthetic control estimators enhances the clarity and concreteness of their arguments. They effectively develop their findings in these two settings first, and thereafter, hinting at the potential extension of the abstract results based on the model-averaging property (MA) and the permutation property (P) to broader settings beyond the two examined scenarios.

While it is yet challenging to assess the significance of this work within the broader NeurIPS readership, I believe the authors make a meaningful contribution to the sub-community of econometrics/causal inference. The paper offers a fresh and clear perspective on understanding the phenomenon of benign overfitting, delivering an important message that reassures the practical utilization of high-dimensional synthetic control, specifically.

**Weaknesses:**

Although this paper has several strengths, there are three concerns/suggestions that could be addressed to further improve its quality.

Firstly, there is a concern about the possibly limited scope of applicability for the presented results. Although the paper establishes theoretical support for the concept of "benign overfitting" in synthetic control estimators and acknowledges the possibility of extending the analysis to a broader class of estimators through the abstract conditions (MA) and (P), it remains unclear how far-reaching this perspective can be and to what extent it can be extended. It would be valuable if the authors could provide a discussion on the expected scope of extensions, limitations, and potential challenges that may arise.

Additionally, in order to strengthen the positioning of the paper, it would be beneficial to provide additional comparisons and contrasts to previous approaches. Given the focus on synthetic control estimators, the authors could comment on the limitations of applying existing results on double descent and benign overfitting in over-parametrized models to the specific context of synthetic control estimators. This would effectively highlight the authors' contributions in this work and emphasize the distinctive features of the techniques employed.

Lastly, while the paper primarily presents a novel theoretical perspective, it would be valuable to complement the theory with a more extensive set of numerical experiments. For instance, conducting an ablation study with synthetic datasets to verify the assumptions and theory, as well as performing experiments with real-world datasets at scale to confirm the descent behavior of the synthetic control method, could strengthen the paper's contributions. By incorporating such experiments, the authors can provide empirical evidence to support their theoretical findings and enhance the practical relevance of the paper.

By addressing these concerns, the authors would be able to further strengthen their contributions in this paper, clarify the scope of their results, highlight its distinctive features, and provide a more comprehensive explanation for the implications of their findings.


**Questions:**

1.  I would like to request the authors to address the three concerns raised in the “Weaknesses” section.

2.  Miscellaneous suggestions:
  (a) It would be clearer if the authors replace the term "full rank" with "full row rank" for clarity, e.g., in lines 127, 171, 174, 191, although it is obvious in the interpolating regime.
  (b) In lines 153-155, the authors state, "In this case, loss continues to decrease throughout, ultimately reaching a minimum that is below the lowest error achieved left of the interpolation threshold." However, this is not easily discernible in Figure 1-(b).

**Limitations:**

The paper suggests several potential avenues for future research, recognizing some limitations of their findings, such as exploring more fundamental conditions beyond (P). However, it could benefit from a more explicit discussion on the limitations and potential negative societal impact of the proposed approaches. It would be valuable for the authors to provide a more detailed acknowledgment of the technical limitations of their methods and offer insights into the potential adverse consequences that may arise when applying these approaches in real-world settings, specifically in the context of synthetic control methods. Nonetheless, it is important to note that given the paper's primary focus on theoretical aspects, a comprehensive and extensive examination of these limitations may not be deemed critical.

---

> ### Author Rebuttal · Authors · 2023-08-10
>
> Thank you for your comments and suggestions.
>
> 1. For the **concerns** you lay out:
>     - We believe that **important next steps on this agenda** include (i) developing low-level sufficient conditions under which more complex synthetic control performs better than less complex synthetic control, (ii) providing tools for valid inference following interpolating linear regression and high-dimensional synthetic control, and (iii) understanding implications for other causal-inference tools that rely on bias–variance trade-offs. Our current analysis is limited since assumptions are on a very high level and the analysis is limited to two specific methods. We believe that deriving inference results in particular may be methodologically challenging, but hope that our analysis can motivate further research in this area.
>     - We now provide **additional comparisons** and results, following the suggestions from other reviewers (see **attached PDF** and our response to reviewer xBLi). In particular, we also consider low-dimensional linear regression, a comparison to the LASSO, and additional evidence for synthetic control with more pre-treatment periods. For our contribution, we believe that the results on synthetic control are new and not previously available in the literature.
>     - Some additional **numerical experiments** are provided in the **attached PDF** and in the response to reviewer xBLi.
>
> 2. Thank you for your **suggestions**, which we will address in our manuscript.
>     - We will replace “full rank” by “full row rank”.
>     - Thank you for your suggestion about Figure 1. We agree that the error in the right tail is comparable to that on the left for the CPS (Figure 1(b)), while it shows a clear improvement for the NSW controls (Figure 1(a)), which we will clarify in the text. In addition, in the graphs in the **attached PDF** we added a zoomed-in panel to facilitate the comparison of performances.

---

> > ### Comment · Reviewer_aRQr · 2023-08-16
> >
> > Thank you very much for the rebuttal and your clarifications for the points raised by myself and other reviewers.  I maintain my initial positive evaluation as it stands, albeit I somewhat agree with the concerns expressed by Reviewer TPer.

---

### Official Review · Reviewer_wCp4 · 2023-07-06

**Soundness:** 3 good
**Presentation:** 3 good
**Contribution:** 2 fair
**Rating:** 6
**Confidence:** 3

**Summary:**

The paper studies the issue of double and single descent for linear regression as well as synthetic control. For the former, empirical motivation is given in the context of predicting wages and theoretical results are given from the perspective of model averaging. For the latter, empirical motivation is given in the context of imputing counterfactual California smoking rates and again theoretical results are given from the model averaging perspective.

**Strengths:**

- To my reading, the most impressive result is Proposition 4, which shows that synthetic control has the model-averaging property. I presume that this is new in the literature but it is not explicitly mentioned in the paper. It would be good to clarify the novelty of Proposition 4 and emphasize its implications.
- The two numerical examples are illustrative and provide very good empirical motivation.


**Weaknesses:**

- To my reading of the literature, one of the most under-explored but central issues in double descent or benign overfitting is the analysis of bias. For example, Tsigler and Bartlett (2023, JMLR) entitled "Benign overfitting in ridge regression" provides sharp bounds for the bias term. The current paper does not provide any result regarding the bias term, which will not be zero with over-parametrization.
- It seems that it is not totally new to combine model averaging with double descent in the literature. For example, see the following quote from Wilson and Izmailov (2020, NeurIPS, page 8) entitled "Bayesian deep learning and a probabilistic perspective of generalization": _"Double descent [e.g., 3] describes generalization error that decreases, increases, and then again decreases, with increases in model flexibility. ... However, our perspective of generalization suggests that performance should monotonically improve as we increase model flexibility when we use Bayesian model averaging with a reasonable prior."_ It would be helpful to provide a more thorough discussion of the literature.


**Questions:**

- Tsigler and Bartlett (2023, JMLR) gives general sufficient conditions under which the optimal regularization parameter is negative. In view of this, I am wondering what would happen if $\eta$ in equation (2) is negative. In other words, would it be possible to study the penalized synthetic control estimator with a negative regularization parameter?
- Model averaging is popular in both statistics and econometrics: e.g., Claeskens and Hjort (2008), Model selection and model averaging, Cambridge University Press; Hansen (2007), Least Squares Model Averaging. Econometrica. Some discussion of related papers on model averaging would be helpful.

**Limitations:**

- Lines 221-223: it is stated that "We believe that these variance and geometric properties of linear regression are well understood in the literature and likely not new, although we are not aware of an explicit statement of the model-averaging connection between more and less complex interpolating linear-regression models." The statement is a bit unclear. It would be better if the literature review is more thoroughly done in the paper.

---

> ### Author Rebuttal · Authors · 2023-08-10
>
> Thank you for your thoughtful comments and suggestions.
>
> 1. For the point about **bias**, we agree that our analysis does not separately provide results for the bias of the estimator, which will generally depend on the relationship of the estimator to the data-generating process. We note, however, that our results on mean-squared error implicitly captures both the variance and bias components of the loss. For example, assumption (P) in _Proposition 5_ also constrains the bias of the estimator, although only implicitly. We agree that a study of bias is a relevant extension to our work, especially for the case of synthetic control.
>
> 2. For **negative regularization**, our results do not directly extend to this case, but this may be an interesting case for further analysis. For the specific case of synthetic control, the optimization problem with a negative cost of complexity is not generally convex any more, solutions are not guaranteed to be unique, and the model-averaging property does not have to hold (although we suspect that this is due to non-uniqueness if ties are broken inconsistently).
>
> 3. On the **relationship to the model-averaging literature**, we agree that it would be worthwhile to add additional references and connections. However, many approaches to model averaging rely on using outcome data to decide on weights to assign to different models, and effectively choose the weights that maximize prediction fit in the training data. This stands in contrast to some of the weights we consider. For example, in the case of interpolating linear regression, weights do not depend on the outcomes, since all component models fit the data perfectly, and we do not use their empirical performance to combine simpler models into a more complex one.
>
> 4. Thank you for pointing out the **relationship to prior work on Bayesian model averaging**, which we will reference. In this context, our approach could be seen as the uninformative limit of a Bayesian regularization approach.

---

> > ### Comment · Reviewer_wCp4 · 2023-08-18
> > **Thanks**
> >
> > I very much appreciate the rebuttal by the authors. I just wanted to re-iterate the importance of a more explicit study of the bias because Assumption (P) is a high-level sufficient condition. It would be good to mention this limitation in the camera-ready version. I changed my rating from 5 to 6 because the authors answered well most of my comments.

---

### Official Review · Reviewer_xBLi · 2023-07-07

**Soundness:** 4 excellent
**Presentation:** 4 excellent
**Contribution:** 3 good
**Rating:** 9
**Confidence:** 4

**Summary:**

This paper examines single and double descent phenomena in two causal inference estimators: high-dimensional linear regression and synthetic control estimators with many controls. To begin, the paper starts with a high-dimensional linear regression problem and illustrates the double descent phenomena using the famous LaLonde dataset. Then, they show that complex models can be seen as the model averaging of simple models when using interpolating linear least squares estimators. Their main contribution is to explore the single and double descent phenomena in synthetic control methods for the first time in the literature. While it is commonly recommended not to include too many control units, the authors found the single descent phenomena: the performance monotonically increased as the number of control units increased. They derive the model-averaging-based risk bounds to explain the single phenomena they observed in the real-world example.

**Strengths:**

**Originality**
To the best of my knowledge, this is the first paper to explore the single and double descent phenomena in causal inference settings. In particular, for synthetic control problems, there are a large number of control units in many important settings. For example, when a company tries to estimate the effect of a certain internal policy on its performance, there are potentially a large number of firms they can use as the donor pool. The conventional recommendation is to focus only on a small number of selected control units, but this paper opens up a new opportunity for researchers to incorporate a large number of control units beyond the number of pre-treatment periods allowed.

**Quality**
This paper provides a very useful insight into the single and double descent phenomena from a model averaging perspective. Providing both intuitive and geometric interpretations of results was helpful.

**Clarity**
This paper is clearly written, and starting with a simpler linear regression as a basis for the synthetic control method was also a very effective presentation.

**Significance**
As mentioned above, it will be significant as this paper will open up a new opportunity for researchers to incorporate a large number of control units beyond the number of pre-treatment periods allowed.

**Weaknesses:**

I do not think there is any obvious weakness in the paper with respect to the goal of the paper. I have some suggestions about making the paper more relevant to realistic causal inference settings, and I list them as clarifying questions below.

**Questions:**

**(1) Comparison to a "reasonable" simple model in a bit more realistic setting**

In many deep-learning settings, it is probably difficult to think about a "reasonable" set of covariates as the variables might have very little substantive meaning in some applications like image detection. But, for many causal inference problems like those examined in this paper, there are many "simple" models that any social scientist would start with. While this paper provides very interesting insight about the single and double descent phenomenon, I think the paper would be even stronger if the authors can demonstrate the idea of double descent and over-parametrization can actually beat simple yet reasonable models. As far as I understand, in both empirical examples (Figure 1 and Figure 4), the baseline model is to include variables "randomly" (random simple models) and then average the performance across such random simple models. And the average of such random simple models can have low performance as it averages over many "random" non-reasonable models. I want to emphasize that the original authors already acknowledged this point in the paper, and I am here to encourage the authors to explicitly include some empirical evidence about this point.

(1.1) LaLonde data

In the LaLonde data example, because many variables are constructed just from eight variables, it is expected that most of the 8000 variables have very low signals and a random subset of such variables can be far from informative variables. **Can the authors include the RMSE based on a simple linear regression that includes the original eight variables additively?** Can overparameterization outperform this baseline model?

(1.2) Synthetic Control Problem

It is interesting to see that it only shows the single descent phenomenon. But I was wondering whether this is due to the very small number of pre-treatment periods (3). Essentially the regime before the interpolation threshold could be too small to show any bias-variance tradeoff in a meaningful sense. **Can the authors include at least ten pre-treatment periods, which is often recommended in practice?** Do we still see the single phenomenon with a larger number of pre-treatment periods?

**(2) Overparametrization vs Regularization**

Another simple benchmark is a classical regularized model. For example, for the Figure 1 problem, **can the over-parameterized model beat a simple Lasso?** For the linear regression problem, the minimal-norm interpolation solution picks a solution that has the minimum norm rather than solving the objective function with a penalty term. **What is the theoretical connection between a classical penalized regression (adding a penalty to the objective function to make a solution unique) and the over-parametrized model (finding all the solutions that make the objective function equal to zero and finding one solution that has the minimum norm)?**

**(3) Inference**

This paper considers the RMSE of causal estimation. But, in many causal inference settings, researchers are also interested in estimating confidence intervals. I am curious to know whether **over-parametrized models** will make inference intractable or can be addressed similarly in the double machine learning or the semiparametric inference literature (for a linear regression case; as far as I know, there is no unified inference framework for the synthetic control method yet).

**(4) Connection to Super Learners**

SuperLearners by van der Laan and his colleagues also use the convex combination of individual prediction methods to improve the performance of the ML prediction. **Is there any connection to SuperLeaners?**

**Limitations:**

They clarify the limitations of their approach.

---

> ### Author Rebuttal · Authors · 2023-08-10
>
> Thank you for your detailed and thoughtful comments and questions.
>
> 1. In response to your question about **reasonable comparison models**, we are reporting additional comparisons:
>    - **For linear regression**, we report a comparison to low-dimensional linear regression using the original covariates for the task of estimating average treatment effects, with a varying size of the target control group (see **attached PDF**).
> Specifically, _Figure R1_ reports the average RMSE for estimating the average treatment effect (ATE) of subsets of varying size $m$ in the experimental NSW sample. (The results reported in _Figure 1 (a)_ correspond to $m=1$, for which the RMSE of imputing control outcomes is the same as the RMSE of estimating the corresponding treatment effect.) In the figure, we now add horizontal lines for the performance of a simple linear regression on the original eight covariates, which shows that the performance of the simple regression is better than the best interpolating regression when $m=1$, but that for $m=50$ and $m=100$ very high-dimensional over-parametrized solutions perform comparably to (and ultimately slightly better than) the simple model.
> We see this as an encouraging finding for interpolating solutions in linear regression, since the complex models do not explicitly use the original regressors (only their binned and interacted versions) but are still able to recover comparable predictive performance. A potential improvement in performance could be to always include the original eight variables in the complex models, without any explicit or implicit regularization, in which case we would expect the comparison to be more advantageous for our complex models.
>    - **For synthetic control**, we also report results for more pre-treatment periods (see **attached PDF**). Specifically, _Figure R2_ reports results analogous to _Figure 4_, but for ten pre-treatment periods. The results are qualitatively similar.
>
> 2. For the **comparison to regularized models**, we report below the performance of LASSO solutions on all covariates in recovering the ATE on the evaluation sample. The relative performance depends on the size of the evaluation sample and the regularization parameter. In this table, every row corresponds to a different sample size of subset on which the ATE is evaluated (see our response above), and each column corresponds to a LASSO penalty parameter or the interpolating solution with all covariates included.
>
>    |$m$   | $\alpha=0.01$ | $\alpha=0.005$ | $\alpha=0.001$ | $\alpha=0.0005$ | Interpolating |
>    |----|---------------|----------------|----------------|-----------------|--------------|
>    |1   | 6.403         | 6.564          | 8.149          | 8.765           | 8.168        |
>    |5   | 3.009         | 3.005          | 3.666          | 3.956           | 3.692        |
>    |10  | 2.293         | 2.226          | 2.588          | 2.752           | 2.512        |
>    |20  | 1.779         | 1.604          | 1.773          | 1.890           | 1.757        |
>    |50  | 1.540         | 1.238          | 1.112          | 1.098           | 1.154        |
>    |100 | 1.319         | 0.985          | 0.730          | 0.699           | 0.771        |
>
>    - The LASSO uses explicit regularization, while our main specification only regularizes among perfectly fitting solutions (for which it chooses the norm-minimal model). The latter approach has the advantage that it does not require us to choose a regularization parameter, but we do not claim that it is optimal. The explicit regularization approach of the LASSO has the advantage that it may further improve performance. Indeed, the finding that very complex, interpolating solutions can perform well and that adding complexity (in the form of additional covariates or donor units) can improve out-of-sample performance still leaves room for improvement by additional, explicit regularization.
>    - The LASSO uses the Taxicab (L1) norm for regularization, while our approach uses the Euclidean (L2) norm. In principle, we could also use the L1 norm for choosing among interpolating solutions, but in that case we may obtain non-unique solutions (at least in some edge cases) since we lose strict convexity. Furthermore, we know from theory that LASSO regression does well in cases with approximately sparse parameters, but may do poorly if most regressors are relevant for prediction.
>
> 3. We agree that **inference** on treatment effects when nuisance components are estimated by very high-dimensional and interpolating models is an important open question. To the degree that inference results for e.g. double machine learning rely on predictive fit or variance properties, our analysis suggests that these results may be feasible even with interpolating solutions, since good out-of-sample performance and low prediction variance do not rely on low dimensionality. Establishing sufficient conditions for valid inference on causal parameters in the presence of interpolating estimation of nuisance components could be a promising direction for future research.
>
> 4. **Super learners** start with a set of candidate learners. The super learner then combines these candidate learners, for example by picking one out of the set, or by picking some linear combination, or a convex combination. Our method is in the spirit of the third flavor of the super learner, if one views the original regressors all as candidate learners. Picking a convex combination as in the super learner then has the averaging property that we exploit. We appreciate you pointing out the connection.

---

> > ### Comment · Reviewer_xBLi · 2023-08-14
> >
> > Thank you so much for the rebuttal and your detailed clarification. These answered my questions well!

---

### Official Review · Reviewer_TPer · 2023-08-01

**Soundness:** 4 excellent
**Presentation:** 3 good
**Contribution:** 2 fair
**Rating:** 4
**Confidence:** 3

**Summary:**

The paper studies overparameterized linear regression in the context of imputing data for estimation of causal effects. This involves both unconstrained regression for imputing wages in the CPS dataset and regression under a simplex constraint on the weights for synthetic controls in the California smoking rates dataset.

Favorable performance is observed for the overparameterized estimators, where the number of donors is larger than the number of samples. The main contribution of the paper is an explanation of this favorable performance. It is shown that minimum norm solutions in overparamterized linear regression can be expressed as weighted averages of least squares solutions under subsets of the donors.
Then it is claimed that such averaging results in better generalization in imputing the missing values.

**Strengths:**

I think that problems regarding the use of overparameterized models in causal inference are very important and interesting.
The paper makes simple yet non-trivial observations about the model-averaging property of min-norm linear regression models when used in causal inference tasks.
It offers some observations that might prove useful in practical considerations on whether one should include as many donors as possible versus selecting donors carefully from a large pool.

Finally, I found the presentation quite simple and fluent. Overall, reading the paper was an enjoyable experience.

**Weaknesses:**

The main weaknesses of the paper in my humble opinion are as follows:
1) I think that the explanation for why model averaging results in better generalization is somewhat lacking. The closest result to a generalization bound is proposition 5, though this only proves the resulting classifier is better than the worst-case classifier. It is quite far from results on benign overfitting such as Liang and Rakhlin, Bartlett et al. and others, which bound the excess risk with respect to the optimal hypothesis and also derive conditions on the data distribution for these results to hold. The results in this paper are not dependent on properties of the data, and the generalization guarantees are quite weak, hence I am not entirely convinced that model averaging is the reason behind the improved generalization, and it might be a red herring. Hence for the unconstrained case, I am not sure whether the model averaging interpretation offers the same understanding of generalization of overparameterized models as previous works such on benign overfitting.
2) Unless there is something I've missed in the paper, the model averaging properties seem to be properties of linear regression (and linear regression under simplex constraints) in general, and they are not specific to causal inference problems. Hence it might be useful to write the paper without the focus on causal inference, to make it appeal to a broader audience which might not be fluent with causal inference techniques. Instead, it would be nice to give the causal inference problems as examples/applications of the more generic result on the types of linear regression.
3) The paper does not discuss the interpretation of regression weights in the overparamterized case. Giving causal interpretations (under certain conditions) to regression weights is one of the major differences in using such models for causal inference, instead of for standard prediction tasks. Hence I'd expect some kind of discussion on such interpretations in this paper, as it focuses on causal estimation.

**Questions:**

1) Are the results data-dependent in some way? Under what formal conditions can we give meaningful bounds in the residual error between the overparameterized solution and the optimal subset of donors (or w.r.t to some baseline donor selection method)?
2) Is there any point in trying to give a causal interpretation to regression coefficients in the method under some standard assumptions? While my intuition is that we should not do that, it is interesting to discuss this explicitly.

**Limitations:**

The authors have properly discussed limitations, there does not seem to be a concern for negative societal impact.

---

> ### Author Rebuttal · Authors · 2023-08-10
>
> Thank you for your thoughtful comments.
>
> 1. The question **under which conditions we obtain better generalization** goes to the heart of the issue. While the model-averaging property holds mechanically for these estimators, the resulting generalization error bounds (like the one in _Proposition 5_) require restrictions on the data-generating process. We agree that understanding lower-level conditions under which model averaging translates into better performance is an important next step. At the same time, we want to clarify that the high-level condition in _Proposition 5_ applies beyond the worst case; it provides a condition under which a classifier of higher complexity outperforms an average classifier of lower complexity, not just a worst-case classifier.
>
> 2. We agree that the question of **how to interpret models** is an interesting one, and we also agree with your assessment that these coefficients should not be given a causal interpretation. This does not preclude a causal use of the resulting predictions, which can be helpful for imputing unrealized potential outcomes.
>
> In addition, we agree that **our results hold beyond the causal-inference context**. At the same time, we believe that our unique contribution is to show that they can be particularly relevant there, since estimation challenges in causal inference have frequently been framed around bias–variance trade-offs. Developments in the literature on double descent and benign overfitting, as well as our own results on synthetic control, have the potential to amend this view in causal inference in particular. In order to further clarify the role in causal inference, we have added some results on using linear regression specifically for the estimation of average treatment effects (see **attached PDF** and our response to reviewer xBLi).

---

> > ### Comment · Reviewer_TPer · 2023-08-13
> > **Response to Rebuttal**
> >
> > Thank you very much for the rebuttal and your clarifications. To be precise about my current point of view on the paper, I think that the problem is interesting and I appreciate the treatment of overparameterization in the context of synthetic controls. I also think that the averaging property is something I did not know about and seems like a nice insight.
> >
> > However, without a strong generalization bound, it is unclear why model averaging as implied by the min-norm interpolator is beneficial for generalization. Concretely, we know there are cases where overfitting is not benign, hence model averaging can also be quite bad. I think that without a characterization of some intuitive data distributions where averaging helps us prove better generalization, the result is nice but lacks meaningful consequences.
> > From reading proposition 5 I was not able to parse such a bound, since the average risk over all possible choices of donors seems like a pretty weak baseline, and I don't have any intuition about types of distributions where condition (P) holds (or even a toy example from which we can gain intuition).
> >
> > So while I am generally positive about this direction of work, the writing, and even like the current results, I think there are missing components that are important to the theory and its connection to practice. The additional experiments conducted with LASSO in response to reviewer xBLI are an interesting start in resolving such issues, but their connection to the theory is still weak and in my view (which may oppose the view of other people involved in the decision here), some more steps are required before publication.

---

> > > ### Author Response · Authors · 2023-08-21
> > >
> > > Thank you for following up on our comments, and for providing further clarifications. We agree that additional theoretical results will be valuable to understand which lower-level conditions are sufficient to guarantee that convex model-averaging translates into improved performance from higher complexity.
> > >
> > > We hope that our manuscript already provides valuable contributions by (1) providing a tuning-free synthetic-control method that applies in the case of many control units, (2) documenting its properties on a real-world dataset, (3) deriving theoretical properties of the estimator, and (4) discussing how these results relate to interpolating regression. We believe that these contributions may motivate and feed into theoretical follow-up work along the lines you suggest. We believe that our results on synthetic control may also be of valuable practical relevance to applied researchers, when there is no strong prior on the importance of individual control units.

---

### Author Rebuttal · Authors · 2023-08-10

We are attaching a one-page PDF with two additional figures: _Figure R1_ presents the performance of linear regression in estimating average treatment effects in samples of varying size, and _Figure R2_ presents the performance of high-dimensional synthetic control with ten pre-treatment periods. Both figures are referenced in our responses to the reviewers, and described in more detail in our response to reviewer xBLi.

---

### Decision · Program_Chairs · 2023-09-21

**Decision:**

Accept (poster)

**Comment:**

The paper builds on recent findings from the ML overparameterization community to give new and important insight into several causal inference methods, including synthetic controls. The paper also includes a sound empirical evaluation, though reviewers had several suggestions as to how to improve those. While there are limitations to the technical novelty (as discussed extensively by the reviewers), it is ultimately more important to open up these connections between two disparate fields, rather than give the best possible theoretical result in a conference paper. The results in the paper are sufficiently interesting on their own, and well-grounded, and I believe this work will generate a good deal of follow up in the community. One missing related paper which should be cited and discussed in the final version is [1].

[1] Kato, Masahiro, and Masaaki Imaizumi. "Benign-Overfitting in Conditional Average Treatment Effect Prediction with Linear Regression." arXiv preprint arXiv:2202.05245 (2022).